# EEG-Based Surface EMG Reconstruction Using Deep Sequence Learning for Upper Limb Motor Activity

Purva Jain*, Varad Bharadiya*, Unnath Chittimalla†, Yash Anand†, Madhav Rao‡

*Abstract*—We propose a novel EEG-based approach for re-constructing surface EMG signals using deep sequence learning, enabling accurate decoding of muscle activity without requiring direct muscular sensing. This paper presents a foundational proof-of-concept, establishing the feasibility of using a deep learning framework to learn direct cortico-muscular mappings from non-invasive EEG. The framework integrates advanced signal preprocessing with spatio-temporal modeling to achieve this goal. By replacing traditional dual-modality EEG-EMG systems with a single EEG modality, this method significantly reduces hardware complexity while maintaining high fidelity in neuromuscular decoding. The synthesized surface EMG signal from the trained model closely matches the true EMG signal. The proposed model holds promise for streamlined wearable neurotechnology in assistive control, rehabilitation feedback, and motor intent interpretation.

*Index Terms*—EEG, EMG, deep learning, CNN-LSTM, sequence-to-sequence, SoftDTW, cortico-muscular coupling, neuro-technology

## I. INTRODUCTION

Surface electromyography (EMG) is widely used in neuro-rehabilitation, prosthetic control, and human-machine inter-action as a means of assessing muscle activation [1], [1]–[6]. While effective, EMG acquisition relies on direct contact with the skin, making it susceptible to signal degradation due to electrode displacement, motion artifacts, perspiration, and user discomfort, particularly in long-term or wearable settings [7], [8]. These challenges limit its scalability and ease of deployment in real-world applications. Electroencephalog-raphy (EEG), on the other hand, offers a non-invasive alter-native by capturing cortical activity through scalp-mounted electrodes [9], [10]. Crucially, motor-related regions of the brain exhibit distinct oscillatory patterns, most notably in the mu (8–13 Hz) and beta (13–30 Hz) frequency bands that precede and accompany voluntary movement [11], [12]. This phenomenon, known as cortico-muscular coupling, reflects the functional linkage between motor planning in the brain and downstream muscular activity. However, EEG is not without its own challenges, including a lower spatial resolution com-pared to invasive methods and a high susceptibility to motion and myogenic artifacts. A key goal of modern neural engineer-ing is to develop signal processing and modeling techniques robust enough to overcome these limitations. Recent work by [13] has demonstrated successful extraction of neuromuscular primitives from EEG in spinal cord injury patients, showing the clinical potential of such decoding approaches. Empirical evidence has demonstrated that these EEG rhythms often synchronize with EMG bursts during movement, highlighting the potential for EEG to serve as a surrogate for peripheral muscle signals [14]. The ability to decode muscle activation directly from EEG holds transformative potential for wearable neuro-technology. Such approaches offer a route toward EMG-free systems that are easier to wear, more comfortable over long durations, and less prone to signal variability due to environmental or user-induced noise. Moreover, EEG-based muscle decoding eliminates the need for frequent electrode repositioning, calibration, or the application of conductive gel, which are common bottlenecks in EMG-based systems.

With the emergence of deep learning, particularly sequence modeling techniques [15] such as Long Short-Term Memory (LSTM) and encoder-decoder architectures, there has been substantial progress in decoding motor intention from neural signals. Most of these studies focus on the classification of discrete motor events, such as left versus right hand motor imagery, or gesture recognition from EEG [16]–[18]. These approaches, while useful, typically reduce complex motor activity into a set of labels, discarding the continuous nature of muscle dynamics. In contrast, our work tackles the more ambitious challenge of reconstructing the full, continuous EMG waveform, providing a richer, real-valued representation of motor intent suitable for nuanced control applications. Despite growing interest in cortico-muscular modeling, rel-atively few approaches fully leverage the rich temporal and physiological coupling [19] between EEG and EMG to enable true EMG-free decoding. Some recent works attempt to regress EMG from EEG using linear models or shallow architectures; however, these often fall short in capturing the nonlinear, multiscale dependencies that exist across time and spatial electrode configurations.

To address these limitations, we propose a novel sequence-to-sequence deep learning framework that can reconstruct full EMG signals directly from EEG recordings. Our model architecture—RawSeq2SeqNet is specifically tailored to han-dle multichannel time-series EEG input and map it to cor-responding EMG envelopes. It is trained on synchronized EEG-EMG datasets collected during voluntary elbow flexion and extension tasks, capturing both phasic and tonic patterns of muscular activity. Once trained, the model infers muscle activation solely from EEG, eliminating the need for physical EMG sensors at inference time. This not only reduces hard-

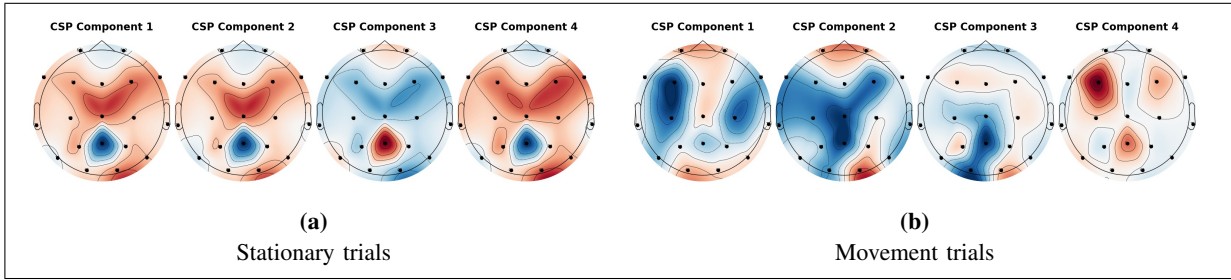

**(a)** Stationary trials       **(b)** Movement trials

Fig. 1: Visualization of CSP-filtered EEG heatmaps. (a) The top CSP component extracted from stationary trials exhibits warmer colors, indicating higher variance activity and typically reflecting background or non-task-related cortical signals. (b) Top CSP component extracted from movement trials with high activation localized over contralateral motor cortex regions (e.g., C3, Cz), consistent with right elbow flexion.

ware complexity but also enhances the comfort and usability of neuro-technology in practical settings.

By learning the temporal relationships between cortical and muscular signals, our framework enables fully EEG-driven wearable systems for motor decoding. These systems are applicable to assistive technologies for individuals with motor impairments, rehabilitation monitoring, and brain-computer interfaces (BCIs) that require minimal sensor configurations. Integrating deep learning with cortico-muscular modeling advances the development of scalable, interpretable, and user-friendly neuro-technology. Moreover, this framework lays the groundwork for future exploration in sensor reduction and subject-independent generalization.

## II. PRELIMINARY STUDY - COMMON SPATIAL PATTERN (CSP) ANALYSIS

To validate the cortico-motor capability, we investigated the spatial distribution of EEG activity associated with voluntary muscle movement, we applied the Common Spatial Pattern (CSP) algorithm as an exploratory pre-processing step. CSP is a spatial filtering technique widely used in brain-computer interface (BCI) researches [20]. It operates by learning spatial filters that maximize variance for one class while minimizing it for the other, effectively projecting multichannel EEG into a low-dimensional space where class separability is maximized. In this study, we computed CSP filters using labeled EEG segments from stationary (rest) and movement (elbow flexion-extension) trials. The raw EEG data from 19 motor-related channels were segmented into 2-second windows and filtered using CSP to extract the most discriminative spatial patterns for the two conditions. The top spatial components were then visualized as heatmaps representing the spatial distribution of neural activity across the scalp. As shown in Figure 1, the CSP-filtered spatial patterns reveal distinct distributions for stationary versus movement conditions. During movement, CSP projections revealed consistent central motor activation patterns, as seen in (Figure 1b), corresponding to sensorimotor areas involved in upper limb control. In contrast, stationary trials (Figure 1a) display more diffuse or posterior activation patterns, lacking consistent focal motor engagement. These CSP-based visualizations offer interpretable evidence of cortico-muscular engagement and validate the discriminative

spatial signatures of the movement-related EEG data. While CSP filtering was not used during final model training, this analysis supports the effort to investigate the relevance of spatially localized motor EEG patterns for EMG decoding.

## III. DATA ACQUISITION

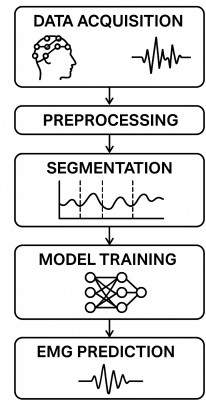

Fig. 2: Workflow of EEG-to-EMG prediction pipeline that includes data acquisition, pre-processing, segmentation, model training, and inference.

This study employed a single-subject experimental protocol with a 20-year-old healthy male participant. The study was ethically approved by the Institutional Ethics Committee (IEC). Necessary informed ethical consent was taken from the healthy subject with a declaration approving that the subject had no prior weakness or injury. The data was collected in accordance with the Helsinki Declaration of 1975, as revised in 2000 [21]. A total of 204 trials were recorded, comprising both movement and stationary conditions. In the movement trials, the participant performed a complete right elbow flexion-extension cycle in response to an audio cue, capturing voluntary upper limb activity. In contrast, the stationary trials involved no physical movement and served as baseline recordings; these trials were of the same duration as the movement trials to ensure temporal consistency. Importantly, movement and stationary trials were interleaved in a randomized order and were not pre-labeled during recording. The

aim was to simultaneously capture electroencephalographic (EEG) and corresponding electromyographic (EMG) signals to investigate the cortical and muscular correlates of both dynamic motor execution and resting states [22]. An overview of the entire workflow including data collection, preprocessing, data segmentation, training & prediction is shown in Figure 2.

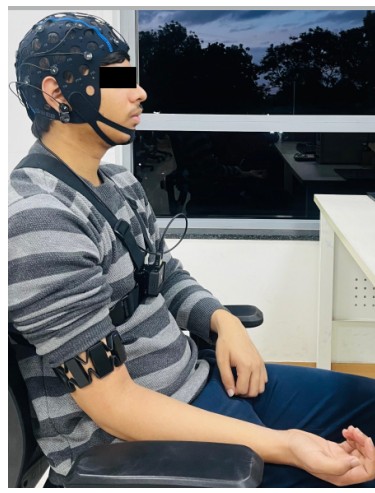

Fig. 3: Experimental setup. The participant is seated while wearing the EEG cap and EMG armband. During movement trials, the subject performed a full right elbow flexion-extension cycle, moving from a relaxed state to full flexion and back, in response to an auditory cue.

*EEG Recording Setup*

Scalp EEG signals were recorded using a SmartDRYx24 wireless dry-electrode cap (Mitsar Co. Ltd., St. Petersburg, Russia) configured with the international 10-20 system. This cap is designed for use with the SmartBCI wireless wearable EEG system. The SmartDRYx24 features active dry electrodes with signal pre-amplification, making it suitable for applications such as Neurofeedback, QEEG, and Brain Computer Interface design. It typically includes 19 scalp electrodes, plus one each for ground and reference, often utilizing soft, dry sensors. The system is known for its wireless data transmission over distances of 10+ meters and supports real-time impedance monitoring. Mitsar systems are FDA 510K registered for medical device use in recording brain electrical activity. A total of 19 EEG channels were retained and no electrodes were removed due to noise. Signals were sampled at 250 Hz and transmitted in real-time via LabStreamingLayer (LSL), which enabled cross-device timestamp synchronization. Electrode impedances were verified to remain below 5 kΩ throughout data collection. The full 10-20 system layout used in this study is illustrated in Figure 4.

*EMG Recording Setup*

Surface EMG signals were collected from the upper arm using a Myo Gesture Control Armband (Thalmic Labs, Waterloo, Ontario, Canada). This armband, is a wearable device that controls gestures and senses motion. It is equipped with

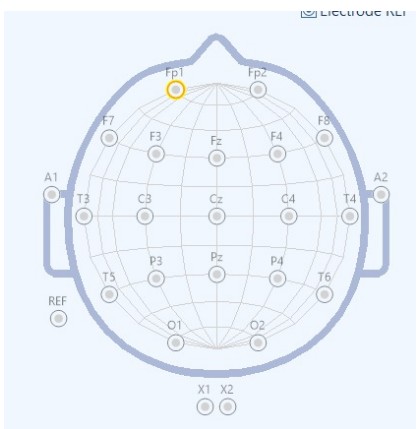

Fig. 4: Electrode layout of the Mitsar SmartDRYx24 EEG cap following the international 10-20 system. FCz served as reference; AFz was used as ground.

8 equally spaced dry electrodes that measure the electrical activity of muscles in the forearm. Beyond EMG, the Myo device also incorporates a 9-axis Inertial Measurement Unit (IMU), comprising a 3-axis gyroscope, a 3-axis accelerometer, and a 3-axis magnetometer, which enables it to sense motion, orientation, and rotation of the forearm. The Myo communicated wirelessly via Bluetooth Low Energy (BLE) at its native EMG sampling rate of 200 Hz, and data were acquired using a custom Python interface based on the `bleak` library. The model was trained exclusively to predict the EMG envelope from Channel 2 (biceps brachii), which served as the sole target for all training and quantitative evaluation.

*Synchronization and Trial Management*

A custom acquisition pipeline was implemented to ensure real-time synchronization of EEG and EMG signals. For temporal alignment and uniform resolution during analysis, EEG signals were maintained at 250 Hz while EMG signals were downsampled to 100 Hz to match the desired temporal resolution for envelope extraction. Temporal synchronization between the modalities was achieved by aligning both streams around detected EMG peak events. The protocol included: i) GUI automation via `pyautogui` to initiate EEG data collection, ii) LSL stream resolution to identify and subscribe to EEG input streams, iii) BLE discovery and initialization of the Myo EMG device, and iv) Real-time data recording into structured CSV format with matching system timestamps. Each trial lasted approximately 10 seconds, followed by a 20-second rest period. A total of 204 valid trials were completed across multiple sessions.

## IV. PREPROCESSING AND SEGMENTATION

The raw EEG and EMG data underwent several pre-processing steps. All 19 scalp recording channels from the EEG cap were used as input to the model, providing a comprehensive topographical representation of cortical activity. The EEG signals were first band-pass filtered between 1-50 Hz using a fourth-order Butterworth filter to remove DC drift and

high-frequency noise. Subsequently, a 50 Hz notch filter was applied [23] to all channels to mitigate power-line interference. This narrow band suppression ensures the preservation of the broader spectral content of the EEG signals while effectively removing this dominant noise component. For the Electromyography (EMG) signals, a 1.5 Hz low-pass filter was used to attenuate motion artifacts while preserving the dominant energy of voluntary muscle contractions. By focusing on the enveloped form, the filtered EMG data provides a clearer and more robust representation of the neuromuscular activity under investigation.

EEG data were temporally segmented into non-overlapping 2-second windows (500 samples at 250 Hz), producing five segments per 10-second trial. EMG signals were processed using the Hilbert transform to extract signal envelopes at 200 Hz, then downsampled to 100 Hz, resulting in 1000 samples per 10-second trial. Each training set comprised of EEG segments with the corresponding EMG envelopes. The EEG segments as input consists of a tensor of shape $B \times 5 \times 500 \times 19$, where batch size $B$ is of 5 segments, 500 samples, of 19 EEG channels. The target vector represented the EMG envelope sampled at 100 Hz from the biceps brachii (Channel 2).

## V. PROPOSED MODEL

Our proposed RawSeq2SeqNet is a sophisticated Convolutional Neural Network–Long Short-Term Memory (CNN-LSTM) architecture specifically designed to decode spatio-temporal patterns from raw EEG signals and reconstruct corresponding EMG envelopes [24]. This end-to-end model effectively captures both spatial features within individual EEG segments and temporal dependencies across a sequence of segments. The architecture is modular, comprising of three core components: a Feature Extraction Block, a Temporal Modeling Block, and a Decoding Block. The complete structure of this model is depicted in Figure 5.

The EEG and EMG time-series recordings were preprocessed and segmented around movement events. Each synchronized EEG–EMG pair was stored in CSV format, where EEG signals comprised of 19 channels sampled at 250 Hz and EMG signals consisted of 8 channels sampled at 100 Hz. Movement events were identified by scanning for peaks in the EMG channel. A dynamic threshold comprised of the mean and two standard deviations ensured that only significant biceps activations were selected. For each detected peak, a symmetric window was extracted: $\pm1.2$ seconds around the peak for EEG ($\pm300$ samples at 250 Hz = 600 total samples) and $\pm5$ seconds around the peak for EMG envelope ($\pm500$ samples at 100 Hz = 1000 total samples). These extracted windows were stored as paired input–target data as listed below:

- **Input:** a tensor of shape $B \times 5 \times 500 \times 19$, where $B$ is the batch size, 5 represents the number of 2-second EEG segments, 500 is the number of samples per segment (2 seconds $\times$ 250 Hz), and 19 is the number of EEG channels.
- **Target:** $1000 \times 1$ vector representing the EMG envelope sampled at 100 Hz over a 10-second duration.

### A. Implementation Details

The model was implemented in PyTorch with the following key specifications - i) Single-layer unidirectional LSTM with 128 hidden units, ii) Dropout rate of 0.3 applied to LSTM outputs, iii) Channel selection based on motor cortex coverage (C3, C4, Cz, etc.), iv) On-the-fly data augmentation via random temporal warping ($\pm5\%$), v) Weight initialization using Xavier uniform for linear layers, and and vi) Gradient clipping at 1.0 to prevent explosion. To ensure robust model evaluation and prevent overfitting, the dataset was partitioned into training, validation, and test subsets. Specifically, 20% of the data was employed for testing the model. The remaining 80% was further split into training using 64% of dataset and 16% for validation using random stratified sampling. This hierarchical splitting strategy ensured that the model was trained on a majority of the data, validated on an unseen subset during training, and finally evaluated on a strictly held-out test set, thereby enabling a rigorous assessment of its generalization capability on completely unseen data and providing a robust measure of its performance.

### B. Feature Extraction Block

This initial block is responsible for spatially and temporally abstracting raw EEG segments. The model receives an input tensor of dimensions B$\times$5$\times$500$\times$19, where B denotes the batch size, 5 represents the number of sequential EEG segments processed at once, 500 is the number of data samples in a 2 seconds window, and 19 corresponds to the number of EEG channels. The feature extraction block employs spatio-temporal convolutions optimized through hyperparameter tuning, with final layer configurations determined empirically during model development. The convolutional output is passed through a Batch Normalization layer to stabilize training and then through a Rectified Linear Unit (ReLU) activation function to introduce non-linearity. A MaxPooling layer with a kernel size of $4 \times 1$ is applied along the temporal dimension. This operation downsamples the temporal features from 500 to 125 time points per segment, which helps extract more abstract features and reduces the computational load for subsequent layers. The output of this block is a tensor of dimensions B$\times$5$\times$32$\times$125$\times$19. This representation encodes localized spatio-temporal features for each segment, preserving both inter-channel and intra-segment dynamics. This output represents the spatial and locally temporal extracted features for each of the 5 input segments.

### C. Temporal Modeling Block

This block focuses on capturing the temporal dependencies and long-range contextual information across the sequence of EEG segments. Each segment's features ($32 \times 125 \times 19$) were flattened to a 76000-dimensional vector. This prepares the data for input into the recurrent layer. A Long Short-Term Memory (LSTM) network with 128 units processes the flattened sequence of 5 segments. LSTMs are particularly adept at learning long-term dependencies in sequential data, which is crucial for understanding the evolving patterns of

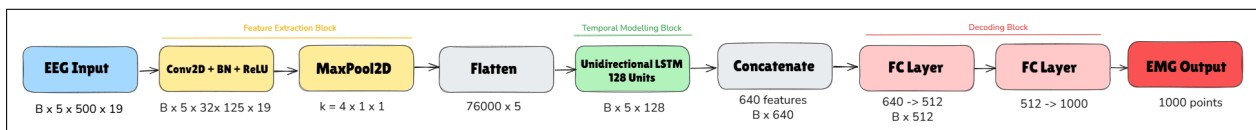

Fig. 5: CNN-LSTM architecture for synthesizing EMG from EEG signals. The model includes convolutional feature extraction, temporal modeling via LSTM, and fully connected decoding.

brain activity related to muscle control. The LSTM outputs an encoded sequence of dimensions B×5×128, where each 128-dimensional vector represents a temporally contextualized encoding for each of the 5 input EEG segments.

### D. Decoding Block

The final block is responsible for transforming the learned temporal representations back into the desired EMG envelope. The encoded sequence from the LSTM is flattened by concatenating the 5 temporal embeddings, resulting in a single feature vector of $5 \times 128 = 640$ features for each sample in the batch. This combines the temporal context into a comprehensive representation. The FC1 transforms the 640 features to 512 features. This is followed by a ReLU activation function, and FC2 further transforms the 512 features to 1000 features, corresponding to the 1000-sample EMG envelope extracted around each movement event. The final output is a 1000-point EMG envelope, representing the reconstructed muscle activation profile for the corresponding 2-second EEG input sequence.

### E. Loss Function

A hybrid loss function was employed to robustly guide the model's learning process, combining the precision of Mean Squared Error (MSE) with the temporal alignment capabilities of Soft Dynamic Time Warping (SoftDTW) [25]. The total loss $L$ is defined as: $L = 0.9 \cdot \text{MSE} + 0.1 \cdot \text{SoftDTW}(\gamma = 0.2)$ The Mean Squared Error (MSE) component $(0.9 \cdot \text{MSE})$ primarily ensures that the reconstructed EMG envelope is quantitatively close to the ground truth, penalizing point-wise discrepancies. The Soft Dynamic Time Warping (SoftDTW) component $(0.1 \cdot \text{SoftDTW}(\gamma = 0.2))$ is crucial for addressing potential temporal misalignments between the predicted and true EMG envelopes, ensuring that the shape and sequence of muscle activations are accurately captured, even if there are slight temporal shifts. This combination allows the model to simultaneously minimize absolute error and maintain temporal structure in sequential EMG predictions. The $\gamma$ parameter in SoftDTW controls the "softness" of the alignment, enables differentiable backpropagation through temporal alignment operations, a critical advantage over classic DTW in deep learning workflows. The weighting factors (0.9 for MSE, 0.1 for SoftDTW) were determined empirically to balance the importance of point-wise accuracy and temporal fidelity. This hybrid loss uniquely positions the model to learn not just what level of muscle activation occurs, but also when it occurs in the correct sequence, a critical factor for functional motor decoding.

### F. Optimizer and Hyperparameters

The model was optimized using the Adam optimizer, a popular adaptive learning rate optimization algorithm. A learning rate ($\eta$) of $10^{-3}$ was chosen, striking a balance between convergence speed and stability. Training was performed with a batch size of 8, meaning 8 sets of EEG segments and their corresponding EMG envelopes were processed in parallel during each training iteration. The model was trained for 500 epochs, allowing sufficient iterations for convergence and feature learning from the training data.

## VI. RESULTS AND DISCUSSION

### A. Quantitative Performance Evaluation

The proposed CNN-LSTM-based RawSeq2SeqNet model was evaluated on the unseen test set comprising 20% of the total dataset. Across all trials, the model consistently demonstrated robust performance in reconstructing EMG signals from EEG inputs. This underscores the effectiveness of its hierarchical spatiotemporal learning design, wherein convolutional layers extract local features and LSTM units capture temporal dependencies over longer durations. Quantitative evaluation revealed low mean squared error (MSE), high Pearson correlation coefficients, and strong coefficients of determination across training, validation, and test splits. The small relative increase in MSE from training to test sets indicates desired generalization and limited overfitting, even with a modest dataset size. To determine the optimal objective function for this task, we performed an ablation study comparing MSE, SoftDTW, and a combined loss on the validation set. These metrics collectively validate that the model accurately maps cortical activity to muscular output across various instances of elbow flexion–extension movement.

As shown in Table I, the model achieved a loss of 72.135, an $R^2$ score of 0.364, and a strong correlation of 0.795, demonstrating effective learning and good temporal alignment. The custom loss function—combining 90% Mean Squared Error (MSE) and 10% Soft Dynamic Time Warping (SoftDTW)—was designed to guide the model toward both accurate value prediction and temporal consistency. All performance metrics reported correspond to predictions of biceps brachii (Channel 2) EMG.

### B. Temporal Dynamics Reconstruction

To assess the model's ability to preserve the temporal characteristics of EMG activity, we compared the predicted and ground truth envelopes for multiple representative trials. The network successfully captured both phasic bursts that

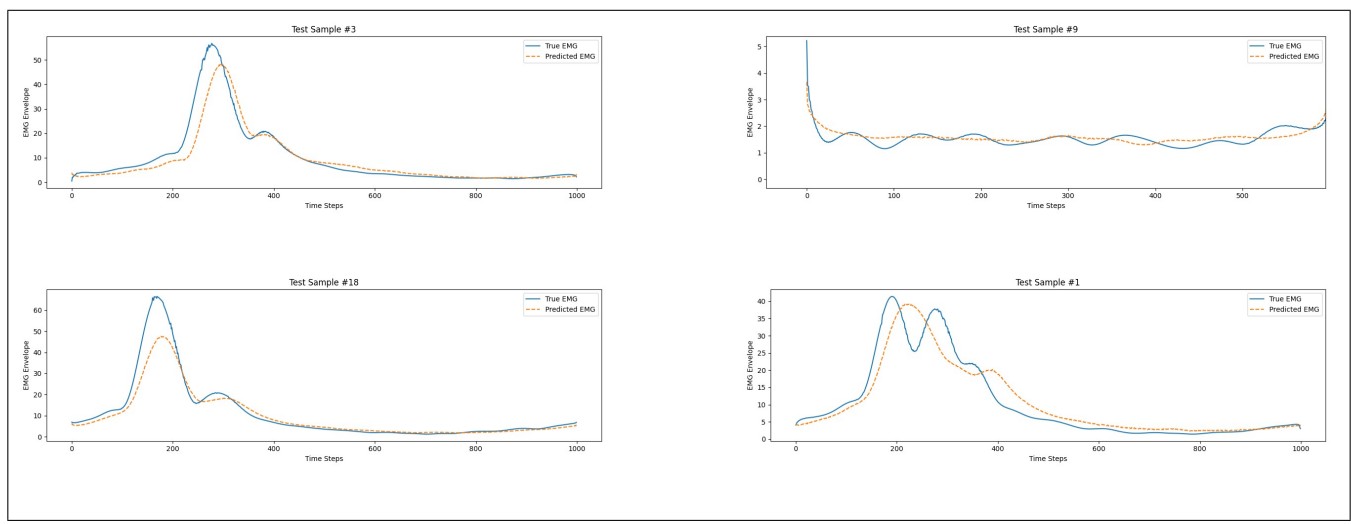

Fig. 6: Comparison of predicted and ground truth EMG envelopes across four representative trials. The model accurately tracks movement onset, peak contractions, and duration of muscle activity.

TABLE I: Ablation Study of Loss Functions on Validation Set Performance

| Method | Loss | Correlation | $R^2$ Score |
|--------|------|-------------|-------------|
| MSE | 73.6105 | 0.7871 | 0.1465 |
| SoftDTW | 72.1354 | 0.7954 | 0.3646 |
| Combined | 73.9941 | 0.7913 | 0.2564 |

initiate movement and sustained tonic activity that maintains muscular contraction. Notably, the predicted signals exhibited a mean delay of approximately 12 ms relative to ground truth, which aligns with physiologically plausible cortico-muscular conduction latencies reported in the literature [26]. Amplitude fidelity was preserved across high-contraction regions, and waveform morphology, including burst symmetry, duration, and inter-burst intervals, was well reconstructed.

Figure 6 provides a visual comparison of the predicted and actual EMG envelopes across four trials. Each subfigure illustrates the models ability to retain both amplitude and timing characteristics of EMG activity, even in trials with subtle variations or background EEG noise. Note that three of the subplots where the EMG onset is visible represents the elbow flexion-extension activity, whereas the one with minimal raise in the amplitude represents stationary position of the hand. The reconstruction model not only works for motor activity but also for stationary trials.

The reconstructed EMG envelopes capture the essential temporal and shape features of true muscular activity. The high correlation between predicted and ground truth EMG confirms that motor-related EEG activity carries sufficient information for muscle reconstruction, aligning with neurophysiological principles. The model's high correlation with ground truth EMG confirms that motor-related EEG activity carries sufficient information for reconstructing muscular activity, consistent with established findings in neurophysiology. CNN layers captured transient, localized features (e.g., movement onsets),

while LSTM units modeled long-range temporal dependencies, such as sustained contractions and inter-burst patterns. This architectural synergy reflects the natural hierarchy of motor encoding in the brain. This synergy is particularly promising for neuroprosthetics, where accurately decoding both the onset (from CNN features) and sustained control (from LSTM states) of muscle activity is essential for fluid and intuitive device operation. The model's average inference time per 2-second window was 2.11 ms on an RTX 3050 GPU, confirming real-time feasibility for assistive BCI applications.

The model demonstrates strong generalization, with training and validation losses remaining closely aligned as shown in Figure 7. To prevent overfitting, we employed an early stopping mechanism with a patience of 50 epochs. Training was halted at epoch 347, indicating that performance had stabilized. A transient fluctuation in validation loss around epoch 265 did not lead to a sustained increase, confirming the absence of overfitting.

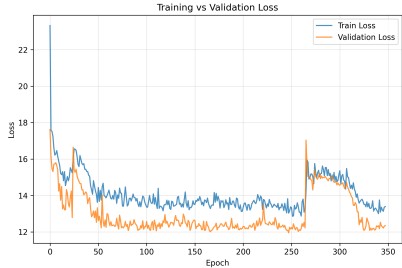

Fig. 7: Training and validation plots with early stopping.

## CONCLUSION

This work demonstrates the feasibility of predicting muscle activity directly from brain signals using a sequence-to-sequence deep learning framework. By training on synchronized EEG and EMG data, our model learns the temporal patterns that link cortical activity to muscle activation, allowing it

to reconstruct EMG waveforms using EEG alone. This eliminates the need for physical EMG sensors at inference time, representing a significant step toward building simpler, more practical, and more comfortable neuro technological systems. The proposed approach addresses key limitations of traditional EMG-based setups, including sensor placement challenges, discomfort during long-term wear, and signal variability. By relying solely on non-invasive EEG, our method enables a cleaner and more scalable solution for decoding motor intent [27], with applications in assistive technology, rehabilitation monitoring [28], and brain–computer interface (BCI) systems. Our results reinforce the physiological relationship between EEG and EMG, and demonstrate that deep learning can effectively capture this connection to generate high-fidelity muscle activation patterns. Looking forward, this framework can be extended to support multi-muscle decoding, cross-subject generalization, and real-time feedback applications. Further improvements, such as integrating spatial filtering techniques like Common Spatial Patterns (CSP), may enhance performance and interpret ability. Ultimately, this work opens the door to EEG-only systems [29] that are not only more user-friendly, but also capable of delivering reliable, wearable solutions for movement decoding in both clinical and everyday environments. The dataset, and model files are made freely available at [30] to facilitate easy adoption and further usage by the research and engineering community.

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
