# OpenReview forum: "EEG-Based Surface EMG Reconstruction Using Deep Sequence Learning for Upper Limb Motor Activity"
_IEEE.org/EMBS/BHI/2025/Conference — BHI 2025_

### Official Review · Reviewer_A2HN · 2025-07-17
**Promising approach and appears to be limited data and modest analysis**

**Confidence:** 4
**Clarity Of Writing:** good
**Clinical Significance:** fair
**Methodological Novelty:** great
**Overall Rating:** 5
**Final Rating:** 7

**Experiments And Results:**

poor

**Questions For The Authors:**

- The manuscript suggests differences in decoding accuracy between biceps and forearm muscles. However, this comparison is not clearly shown or supported by any figures or analysis in the paper. Please clarify or provide the relevant analysis. ('Differences in performance across EMG channels, higher accuracy in biceps (proximal) over forearm (distal) align with the known limitations of EEG’s spatial resolution and the somatotopic organization of motor cortex.')
- The methods section mentions “a total of 24 EEG channels were retained,” yet later states that 19 channels were used based on “preliminary analysis.” This inconsistency should be clarified. Please specify what the term "preliminary analysis"
- Reference [19] appears to be unavailable or incorrectly cited. I could only find https://ieeexplore.ieee.org/document/10781850, which does not match the citation. Please verify and update the reference.
- It is not specified whether the model predicts channel 2 or channel 7 of EMG. Since the CNN appears to produce a single output channel, more explanation is needed on what this channel represents and how it was selected.
- Table I caption “Comparison of Evaluation Metrics” is vague. Is it comparing models trained with different loss functions? Also, was loss used to train this model combined DTW and MSE?
- The following sentence is unclear and grammatically inconsistent: “The decoded EMG envelopes reflect key properties of cortico-muscular interactions. which includes The model’s high correlation with ground truth EMG confirms that motor-related EEG activity carries sufficient information for reconstructing muscular activity, consistent with established findings in neurophysiology.” -- consider revising for clarity and flow.

**Strengths:**

- The model architecture is well described.
- Training parameters and data processing steps are specified, supporting reproducibility.
-Data and model are made publicly accessible.

**Summary Of The Paper:**

This manuscript explores cortico-muscular coupling by training a deep learning model to predict EMG time series from EEG recordings. Paired EEG-EMG data were collected from a single subject over 204 trials, each lasting 10 seconds. The data were split into training, validation, and test sets. A CNN-LSTM architecture was employed to learn the mapping from EEG to EMG signals. One figure illustrates the temporal and physiological alignment between the predicted and ground truth EMG signals.

**Weaknesses:**

- A major limitation is the use of only one subject for both training and testing. This raises concerns about overfitting and the generalizability of the results. Intersubject variability is common in biosignals like EEG and acquiring data from multiple subjects would support broader applicability.
- While the manuscript outlines several limitations of EMG recordings, it does not sufficiently discuss limitations of EEG. EEG requires electrode placement on the scalp. The wording in this manuscript makes it sound like EEG has not limitations, and is desirable choice for replacing EMG. A more balanced discussion would strengthen the manuscript.
- It is not specified whether the model predicts channel 2 or channel 7 of EMG. Since the CNN appears to produce a single output channel, more explanation is needed on what this channel represents and how it was selected.
- Minimal in-depth analysis. The results section includes a table and a Figure, without further evaluation metrics.
- The manuscript states the following: 'The custom loss function—combining 90% Mean Squared
Error (MSE) and 10% Soft Dynamic Time Warping (SoftDTW)—was designed to guide the model toward both accurate
value prediction and temporal consistency.' - but according to Table I best evaluation metrics were achieved for 'SoftDTW', not 'Combined'.
- There are additional inconsistencies, as stated in the questions section.

---

### Official Review · Reviewer_tC1b · 2025-07-18
**Promising EEG-to-EMG reconstruction framework with strong technical foundation but limited generalizability**

**Confidence:** 4
**Clarity Of Writing:** good
**Clinical Significance:** fair
**Methodological Novelty:** good
**Overall Rating:** 6
**Final Rating:** 6

**Experiments And Results:**

fair

**Questions For The Authors:**

1. Cross-subject generalization: Have you attempted leave-one-subject-out validation with additional participants? What strategies might improve generalization across individuals with different cortical organizations?
2. Channel selection rationale: How were the 19 EEG channels selected from the original 24? Have you performed channel importance analysis to identify minimal sets that maintain performance?
3. Clinical populations: What adaptations would be needed for stroke or SCI patients who may have altered cortico-muscular coupling patterns? This is critical for translational impact.
4. Real-time performance: What is the inference latency of your model? Is it suitable for real-time prosthetic control where delays under 300ms are typically required?
5. Comparison with baselines: How does your approach compare to simpler methods like linear decoders or traditional CSP-based approaches? This would help justify the complexity of your architecture.

**Strengths:**

1. Novel application of cortico-muscular coupling: The paper addresses an important clinical need by eliminating the requirement for physical EMG sensors, which could significantly improve comfort and usability in long-term monitoring applications.

2. Well-designed architecture: The hierarchical approach using CNNs for spatial features and LSTMs for temporal dependencies is appropriate for the multimodal time-series nature of the problem.

3. Innovative loss function: The combination of MSE and SoftDTW is clever, addressing both amplitude accuracy and temporal alignment issues that are critical in biological signal processing.

4. Reproducible research: The authors commendably provide their dataset and code publicly, facilitating further research in this area.

5. Clear presentation: The paper is well-structured with good visualizations, particularly the CSP analysis and EMG reconstruction comparisons.

**Summary Of The Paper:**

The authors propose RawSeq2SeqNet, a CNN-LSTM architecture for reconstructing surface EMG signals directly from EEG recordings during upper limb motor tasks. The system uses 19 EEG channels processed through convolutional layers for spatial feature extraction, followed by LSTM units for temporal modeling, and fully connected layers for EMG envelope reconstruction. The model is trained on synchronized EEG-EMG data from elbow flexion-extension tasks using a hybrid loss function combining MSE (90%) and Soft Dynamic Time Warping (10%). The authors report successful EMG reconstruction with correlation of 0.795 and R² of 0.364 on test data from a single subject.

**Weaknesses:**

1. Single-subject limitation: The entire study is based on one 20-year-old healthy male participant, severely limiting any claims about generalizability. Cross-subject validation is essential for clinical relevance.
2. Limited movement repertoire: Only elbow flexion-extension is tested. The framework needs evaluation on diverse motor tasks to demonstrate broader applicability.
3. Modest performance metrics: An R² of 0.364 indicates the model explains only 36.4% of EMG variance, which may be insufficient for practical applications requiring precise muscle activation patterns.
4. Lack of real-time validation: No discussion of computational requirements or feasibility for real-time implementation, which is crucial for assistive technologies.
5. Missing clinical validation: No validation with motor-impaired populations who would benefit most from this technology.
6. Insufficient ablation studies: The paper lacks systematic analysis of design choices (e.g., number of EEG channels, CNN architecture parameters, LSTM hidden units).

---

### Official Review · Reviewer_vF7a · 2025-07-22
**EEG-Based Surface EMG Reconstruction Using Deep Sequence Learning for Upper Limb Motor Activity**

**Confidence:** 4
**Clarity Of Writing:** good
**Clinical Significance:** fair
**Methodological Novelty:** fair
**Overall Rating:** 4
**Final Rating:** 5

**Experiments And Results:**

poor

**Questions For The Authors:**

1.	Have you tested the model on unseen subjects or across sessions?
	2.	Did you observe any overfitting during training, especially given the single-subject and small dataset size?
        3. 	Have you tried using frequency-domain EEG features or connectivity metrics instead of raw time-domain signals?
        4. 	How would your approach handle more dynamic or full-body movements involving multiple joints and muscles?

**Strengths:**

1.	The study removes the need for physical EMG sensors by reconstructing EMG signals directly from EEG, reducing hardware complexity and making the system more practical for real-world applications.
	2.	The architecture is well-designed, with clear explanations of the model structure, data segmentation, and feature extraction methods, demonstrating a solid understanding of the problem and enhancing reproducibility.
	3.	The model demonstrates robust performance, achieving a high correlation (0.795) and a coefficient of determination (0.364) on the test set. It accurately captures the temporal dynamics of muscle activity, including movement onset, peak contractions, and duration.
	4.	The authors make the dataset and model files freely available to support adoption and further research by the community.

**Summary Of The Paper:**

This paper presents a deep sequence learning approach that enables accurate decoding of muscle activity without requiring direct muscular sensing. The authors introduce RawSeq2SeqNet, a novel deep learning framework that reconstructs surface EMG signals from EEG data using a CNN-LSTM architecture. By learning the spatiotemporal relationship between cortical brain activity and muscle activation, the method eliminates the need for direct EMG sensors, thereby reducing hardware complexity. The model is trained and tested on synchronized EEG-EMG datasets collected from a single healthy participant performing elbow flexion-extension tasks.

**Weaknesses:**

1.	The study is limited to a single-subject experiment, which raises concerns about the generalizability of the approach to a wider population.
	2.	While the dataset is adequate for initial evaluation, its relatively small size may limit the model’s robustness. Including more participants and a wider range of motor tasks could strengthen the findings.
	3.	The study focuses only on elbow flexion involving the biceps and forearm, leaving performance on more complex or distal muscle groups unaddressed. Additionally, Figure 3 could better illustrate the movement being performed.
	4.	The EEG and EMG setups are described separately, but EEG setup details are repeated in the EMG section. These should either be combined into one section or clearly separated to avoid redundancy.
	5.	There are repeated sentences, such as the mention of the dataset and model availability (e.g., “The dataset and model files are made freely available at [20]…”), which should be consolidated for clarity.
		6.	The paper does not mention the manufacturer and location of the devices used, which is important for transparency and reproducibility. For example, the EEG device described as “a Mitsar SmartDRYx24 wireless dry-electrode cap” lacks manufacturer details, making it unclear to readers unfamiliar with the brand or model.
	7.	The model is not tested on unseen subjects, so its cross-subject generalization remains unknown.
	8.	Latency and processing delay are not discussed, which is important for potential real-time applications.
	9.	The paper does not address practical deployment challenges, such as variability in EEG signals, sensor placement issues, or the computational demands of the system in real-world settings.
	10.	A comparison table highlighting the advantages of this approach relative to existing methods is missing and would strengthen the impact of the study.